# Effects of Red and Infrared Laser Therapy in Patients with Tinnitus: A Double-Blind, Clinical, Randomized Controlled Study Combining Light with Ultrasound, Drugs and Vacuum Therapy

**DOI:** 10.3390/jpm13040581

**Published:** 2023-03-26

**Authors:** Vitor Hugo Panhóca, Antônio Eduardo de Aquino Junior, Viviane Brocca de Souza, Simone Aparecida Ferreira, Lais Tatiane Ferreira, Karina Jullienne de Oliveira Souza, Patricia Eriko Tamae, Marcelo Saito Nogueira, Vanderlei Salvador Bagnato

**Affiliations:** 1Biophotonics Laboratory, Institute of Physics of São Carlos, University of São Paulo, São Carlos 13563-120, SP, Brazil; vhpanhoca@ifsc.usp.br (V.H.P.); antoniodeaquinojr@ifsc.usp.br (A.E.d.A.J.); vivianesouzasabino@gmail.com (V.B.d.S.); tuttyferreira@uol.com.br (S.A.F.); laistatiane@hotmail.com (L.T.F.); vander@ifsc.usp.br (V.S.B.); 2Institute of Physics of São Carlos, University of São Paulo, São Carlos 13563-120, SP, Brazil; 3Brotherhood of the Holy House of Mercy of São Carlos, São Carlos 13561-060, SP, Brazil; 4Central Paulista University Center—UNICEP, São Carlos 13563-470, SP, Brazil; tamaepe@hotmail.com; 5Integrated Therapy Center, Londrina 86055-240, PR, Brazil; karina@centrointegradodeterapias.com.br; 6Tyndall National Institute, University College Cork, T12 R5CP Cork, Ireland

**Keywords:** low-level laser therapy, vacuum therapy, ultrasound, tinnitus, photobiomodulation, phototherapy, laser puncture, Ginkgo Biloba, flunarizine dihydrochloride, personalized medicine

## Abstract

Background: tinnitus is a symptom with no specific cause known to date, and there are no associated pharmacogenomics of hearing disorders and no FDA-approved drugs for tinnitus treatment. The effectiveness of drug treatments is not reproducible on idiopathic patients and inexistent in refractory patients. Personalized treatments for these patients are a great clinical need. Our study investigated the outcome of potential alternative and complementary treatment modalities for idiopathic and refractory tinnitus patients. Methods: we were the first to evaluate the tinnitus handicap inventory (THI) score changes over the course of treatment up to 15 days after complete cessation of treatment for novel transmeatal low-level laser therapy (LLLT) modalities using light alone, as well as LLLT combined with vacuum therapy (VT), ultrasound (US), Ginkgo biloba (GB) and flunarizine dihydrochloride (FD), while also comparing all treatment outcomes with laser puncture (LP), FD alone and GB alone. Results: a positive treatment outcome (superior to a placebo effect) was achieved by using either LP or transmeatal LLLT, whereas short-term antagonistic effects of VT, US, GB and FD when combined with LLLT. For transmeatal LLLT, an improvement in the treatment outcome was observed by increasing the irradiation time from 6 min to 15 min (with 100-mW of applied laser power at 660 nm). Finally, a lasting therapeutic effect higher than the placebo was observed at 15 days after treatment upon combining LLLT with VT, GB or by using FD alone, by using the transmeatal LLLT alone or by using LP. Conclusions: LP and Transmeatal LLLT can be promising alternative treatments for idiopathic and refractory tinnitus patients. Future studies should investigate the long-term effects of LLLT in tinnitus patients, as well as the dosimetry and wavelength of transmeatal LLLT.

## 1. Introduction

Tinnitus is a symptom of controversial origin. Previous studies have associated the cause of tinnitus as insufficient peripheral irrigation in the tissues close to the labyrinth (internal auditory apparatus) [1]. The main theories suggest increased tinnitus activity in the dorsal cochlear nucleus and inferior colliculus of the brainstem [2]. The same theories indicate that the outer hair cells (OHC) are damaged first and the inner hair cells (IHC) next. Cochlear damage occurs after acoustic trauma altering the spontaneous firing rates of neurons in the dorsal cochlear nucleus (DCN). Another pathophysiological mechanism causing of tinnitus may be a thalamocortical lesion with decreased auditory stimulation [3]. Since tinnitus causes cannot be confirmed for every patient, the consequence is impairment of treatment planning and patient prognosis.

Studies supporting the lack of peripheral irrigation in the inner ear as the cause of tinnitus also associate this lack as the cause of vertigo and hearing loss. In this case, sensorineural hearing loss caused by microcirculatory insufficiency may result from vascular occlusion due to embolism, hemorrhage or vasospasm, diabetes mellitus (DM) or arterial hypertension (AH) [4]. It is worth noting that DM is the condition most commonly associated with hearing disorders compared with glucose-metabolic-related disorders [4]. Additionally, DM patients often present symptoms such as dizziness, tinnitus and hearing loss [4]. 

Currently, there is no recommended treatment for tinnitus. Medicines such as sedatives, antihistamines, antidepressants, local anesthetics and antipsychotics are prescribed for treatment, with different outcomes [5]. These medicines include flunarizine dihydrochloride and Ginkgo biloba. Flunarizine dihydrochloride is a vasodilating drug used for balance disorders (such as vertigo) caused by functional disorders of the vestibular system, dizziness and labyrinth diseases as well as for memory disorders and difficulty concentrating [6]. Ginkgo biloba is used for vertigo and tinnitus resulting from peripheral circulatory disorders and cerebrovascular insufficiency [7]. In general, medicines mitigate patient symptoms in a short-term way and can only be prescribed for continuous use at limited dosages, at times insufficient to completely contain patient symptoms. Continuous use for long periods may also cause side effects. 

Since previous studies have suggested microcirculatory insufficiency as one of the main causes of tinnitus, alternative therapies to increase regional blood flow have been proposed. One of these therapies is low-level laser therapy (LLLT), which has been introduced as an alternative treatment for cochlear dysfunction, tinnitus [8] and muscle and skeletal disorders. LLLT is typically used to promote the modulation of inflammatory response along with analgesic, angiogenic and healing effects based on light absorption by biological tissues and associated photobiomodulation through metabolic activation, the stimulation of the cellular respiratory chain in mitochondria, increasing vascularization and the formation of fibroblasts [9]. A specific type of LLLT stimulating the internal meridians in traditional Chinese medicine is known as laser puncture (also known as laser acupuncture). Since laser puncture illuminates points of acupuncture and is not directed to ear tissues, positive effects are associated with giving the body its energetic balance through the stimulation of the Qi (energy) that runs through its meridians. Another potentially alternative treatment for tinnitus is vacuum therapy (also known as cupping therapy), which is a non-invasive procedure that uses suction cups and negative pressure to increase peripheral blood supply by sucking the skin and subcutaneous tissues. This increase promotes local angiogenesis, leading to an increase in oxygen and nutrients in the applied region, causing anti-inflammatory photobiomodulation and tissue repair [10,11,12,13]. Similarly, ultrasound (US) therapy could be used as a non-invasive treatment to decrease symptoms of inflammation and promote tissue regeneration by applying mechanical waves (sound) to induce cavitation and indirectly heat deep tissues such as tendons, muscles and joints [14]. 

According to the systematic reviews of Chen et al. [15] and Talluri et al. [16] on randomized controlled trials (RCTs) for the LLLT of tinnitus patients, no statistically significant differences have been found in tinnitus handicap inventory (THI) scores nor the visual analog scale (VAS), verbal rating scale (VRS) and numeric rating scale (NRS) scores between placebo and treated groups. No differences have been identified independent of the number of irradiation sessions (varying from three to twenty sessions), the wavelength used (532-nm, 635-nm, 650-nm, 810-nm, 830-nm or 1064-nm), underlying the comorbidity of recruited patients (i.e., sensorineural hearing loss, noise-induced hearing loss or idiopathic) or the type of treatment application. Treatment has also been applied either transmeatally (light delivered at the external auditory meatus) or cochearly (post-auricular application). Since there is no scientific evidence regarding the scale of LLLT’s effect, the authors of systematic reviews [15,16] concluded that large-scale and well-designed future studies are still necessary to evaluate the efficacy of LLLT for the management of tinnitus. With the above in mind, there is an unmet clinical need to propose complementary and alternative treatments personalized to tinnitus patients’ who are either idiopathic or refractory to conventional therapy. The aim of this study was to investigate and propose personalized treatment options based on the effects of transmeatal or cochlear LLLT combined with vacuum therapy, US or drug therapy.

To propose complementary and alternative treatments, none of the previously reviewed studies by Chen et al. [15] and Talluri et al. [16] reported LLLT using powers >5 mW for wavelengths close to 650 nm, and four studies used transmeatal LLLT with wavelengths of ≥810 nm (i.e., 810-nm, 830-nm or 1064-nm). In addition, these four transmeatal LLLT studies used either 100 mW of power or 250 mW of power for 10 sessions over 2 weeks, 450 mW for 12 sessions over 4 weeks or 50 mW for 15 sessions over 3 weeks. Our study investigated the treatment outcome of cochlear LLLT with 100 mW of power (each illuminated spot; total 600mW) at 808-nm and/or 660-nm wavelengths, as well as transmeatal LLLT with 100 mW power at 660 nm for 8 sessions over 4 weeks. Not only does our study report the outcome of a new LLLT modality, but it also evaluates potentially synergistic and antagonist effects due to combinations with vacuum therapy, ultrasound (US) and Ginkgo biloba, while also comparing all treatment outcomes with laser puncture. To the best of our knowledge, our study is the first to (1) show a treatment effect of transmeatal LLLT that is superior than placebos based on THI scores, (2) investigate the synergistic and antagonistic effects of LLLT combined with vacuum therapy, ultrasound (US), Ginkgo biloba and flunarizine di-hydrochloride, (3) monitor short-term effects of nine treatment modalities for tinnitus in both immediately after the treatment and 15 days after its complete suspension and (4) suggest LLLT protocols with positive treatment outcomes for idiopathic and refractory patients.

Since physiotherapy and complementary alternative therapies, including the application of vacuum therapy or US with LLLT, could treat muscle and temporomandibular joint (TMJ) dysfunctions and promote angiogenesis to accelerate healing and scarring, we have hypothesized that a similar synergistic effect of combined applications could increase the therapeutic speed and efficiency by promoting tissue healing and deinflammatory photobiomodulation of the tissues of the inner ear or of the central auditory pathways. In this study, we hypothesized that the application of vacuum therapy or US could enhance the peripheral and regional blood circulation of the hearing system. By performing sonic activation of the tissues, periauricular US application could potentially not only improve the circulatory system, but also promote an anti-inflammatory effect to relieve tinnitus symptoms. To objectively assess potentially synergistic and antagonistic effects for each combination of explored therapeutic modalities, we have analyzed the potential improvement in patient quality of life based on the improvement of THI scores primarily based on eleven questions evaluating functional improvement, nine questions evaluating emotional improvement and five questions evaluating severe conditions resulting from tinnitus. Based on the THI scores, we quantified and compared the treatment outcome after 15 days of complete cessation of treatment ($TO_{15})$, the 15-day lasting therapeutic effect ($LTE_{15}$) and the treatment outcome immediately after the finishing treatments ($TO_{0}$). By statistically comparing the $TO_{0}$, $TO_{15}$ and $LTE_{15}$ of LLLT combined with vacuum therapy, US or drug therapy in 10–11 patients per group, our study provided an objective assessment of the most successful tinnitus treatments in idiopathic and refractory patients as a basis for studies with larger patient populations.

## 2. Materials and Methods

### 2.1. Clinical Research Ethics

The clinical procedure was performed at the outpatient clinic of Santa Casa de Misericórdia de São Carlos in partnership with the Biophotonics Laboratory of the Physics Institute of São Carlos (IFSC) at the University of São Paulo (USP). A total of 107 female and male volunteers aged between 18 and 65 years and with both idiopathic and refractory tinnitus complaints were selected. The demographics of patients receiving treatment are shown in Table 1. Volunteers were recruited through the websites of CEPOF-IFSC (http://cepof.ifsc.usp.br/, accessed on 14 April 2021) and USP (http://www.saocarlos.usp.br/, accessed on 14 April 2021), among others. They were recruited through newsletters calling on volunteer patients to treat patients with uni- or bi-lateral tinnitus. Volunteers who had endocrine diseases, diabetes, blood pressure problems, cancer and obesity were excluded from the research.

This double-blind, controlled, randomized clinical study was approved by the Ethics Committee of the Brotherhood of the Holy House of Mercy of São Carlos, São Carlos, SP, Brazil. (Approval No: 4505913-01/21/2021). All participants handed in written consent and agreed to take part in the study. This study is registered in the Brazilian Clinical Trials Registry/Registro Brazilian of Essays Clinicians (ReBEC) under the clinical trial registration number 10859 with the identifiers:CAAE issuing body: Plataforma Brasil (40341220.0.0000.8148)WHO International Clinical Trials Registry platform: UTN code U1111-1267-9380Approval number of the clinical research ethics committee (from Brotherhood of the Holy House of Mercy of São Carlos, São Carlos, SP, Brazil): 4505913.

### 2.2. Study Protocol

Patients were randomly divided into 10 study groups (Figure 1) treated twice a week over 4 weeks (total of 8 sessions):

Group 1 (n = 11): negative control group (no treatment applied). The LLLT device was turned off before positioning the device to apply auriculotherapy for 6 min. After the end of the research and clinical evaluation, the patients in Group 1 (control) were invited to undergo treatment LLLT and laser for auriculotherapy. Patients accepting the invitation were reallocated to G3.

Group 2 (n = 11): group treated with vacuum therapy and LLLT. The laser power was set at 100 mW for each laser spot. Laser was applied at 660 nm (3 spots; red light) and 808 nm (3 spots; infrared light) wavelengths. Vacuum was applied for 3 min around the ear in the MP7 mode (continuous) and at a pressure of −120 mbar (Figure 2). LLLT and vacuumtherapy were applied transcochlearly (i.e., in the post-auricular region) by using the same device (Vacumlaser–MMOPTICS, São Carlos–SP) configured appropriately for different settings.

Group 3 (n = 11): group treated with vacuum therapy, transcochlear LLLT for 6 min and flunarizine dihydrochloride as a vasodilating drug. The same LLLT protocol as Group 2 was associated with Vertix^®^ (flunarizine dihydrochloride) with a dosage of 10 mg/day of flunarizine (equivalent to 1 flunarizine dihydrochloride tablet or 11.80 mg of flunarizine dihydrochloride). One Vertix^®^ tablet per night was prescribed until the 8th session (last session) for each volunteer. The full duration of treatment was at the discretion of the doctor and, depending on the indication, can vary from 2 weeks to several months.

Group 4 (n = 11): group treated with only Vertix^®^, 10 mg/day. Pseudoapplication of LLLT was performed by positioning the LLLT auriculotherapy device turned off in the ear canal for 6 min.

Group 5 (n = 10): group treated with LLLT at 660 nm and 808 nm and treated with US at 1.0 MHz frequency, 1 W/cm^2^ intensity and 50% pulsed duty cycle for 180 s in the pre-auricular region. When combining US and LLLT, patients were treated with a device enabling the use of US and LLLT simultaneously. We applied LLLT and US transcochlearly (i.e., in the post-auricular region) with 100 mW of laser power, 1.76 mm^2^ laser spot area and 1.6 cm^2^ of US effective radiation area (ERA). We also applied a transparent water-based gel on the tissue to enhance the transmission of the US. Treatment was carried out by performing slow, circular, smooth movements on both right and left sides of the patient’s face.

Group 6 (n = 10): group treated with a prototype device for LLLT auriculotherapy at a 660-nm wavelength (Red light) for 6 min. The laser power was fixed at 100 mW and directly applied to the ear canal. For auriculotherapy, a prototype LLLT device was developed at the Technological Support Laboratory of the IFSC-USP, with a 1.5-mm spot diameter to be directly applied to the ear canal (Figure 3).

Group 7 (n = 11): group treated with a prototype LLLT auriculotherapy device for 6 min combined with 120 mg (1 tablet) of Ginkgo biloba at bedtime. Application of LLLT auriculotherapy followed the protocol described for Group 6.

Group 8 (n = 10): group treated with only 120 mg (1 tablet) of Ginkgo biloba per night at bedtime for 30 days. Pseudoapplication of LLLT was performed by positioning the LLLT auriculotherapy device turned off in the ear canal for 6 min. Ginkgo biloba tablets (120 mg) were coated and had to be swallowed whole.

Group 9 (n = 11): group treated with a prototype LLLT device for auriculotherapy at a 660-nm wavelength (Red light) for 15 min. The laser power was fixed at 100 mW and directly applied to the ear canal.

Group 10 (n = 11): group treated with laser puncture with 4J in a total of 23 acupoints not restricted to the auricular region. These acupoints comprised of 8 bilateral distal acupoints (16 individual points), ST36, SP6, F3, R3 and VB43, C7, LI4, TA5, 3 acupoints at bilateral locations used ID19, VB2 and TA17 (6 individual points) and a single Yin Yang point was also used. The treatment involving these acupoints was proposed to restore the energy balance and Yin and Yang balance of patients, normalizing the flow of Qi (energy) and blood, nourishing deficiencies, reducing excesses and reassuring Shen (mind).

### 2.3. Clinical Evaluation

Clinical evaluation was carried out in the pre- (T0) and post-treatment period (T1; after 8th session) and 15 days after the treatment has been completely suspended (T2). This evaluation consisted of an assessment using the tinnitus handicap inventory (THI) questionnaire developed by Newman et al., 1996. [17,18]. Clinical evaluation was performed after that without informing the data analyst of which data belonged to which study group.

The questionnaire was applied pre- and post-application in each group. The THI questionnaire is composed of twenty-five questions and grouped into three subscales. The functional subscale was composed of 11 questions to assess the limitations that tinnitus causes in the mental, social, occupational and physical areas. The emotional subscale was composed of 9 questions to evaluate affective responses to tinnitus (anger, frustration, irritability and depression). Finally, the catastrophic subscale was composed of 5 questions to investigate the most severe reactions resulting from tinnitus (such as despair, loss of control, inability to face problems, inability to escape tinnitus and fear of having a serious illness). The allowed answers were:“yes”, equivalent to four points in the evaluation scores“sometimes”, equivalent to two pointsand “no”, equivalent to zero points.

The score ranges from 0 to 100 points, and the closer to 100, the greater the disadvantage caused by tinnitus in the patient’s life [18].

### 2.4. Data Analysis

Differences in the initial THI scores of each study group indicate the positive or negative treatment outcome (TO) or lasting therapeutic effect (LTE) of each treatment modality evaluated in our study. However, a reduction in THI scores over the course of treatment means a positive outcome, since this reduction shows that the patient discomfort due to tinnitus (quantified using the THI) has decreased. Instead of showing a negative metric (such as negative THI score changes over time) for positive treatment effects, we illustrated our results by using metrics proportional to the outcome of each treatment.

To do this, we first have assessed the treatment outcome after 15 days of complete cessation of treatment ($TO_{15})$ by calculating the average of differences between THI score of T0 and T2 normalized by the initial THI score at T0 for each patient:(1)$$TO_{15}\;of\;Gn=-\frac{1}{m}\times \overset{\#\;of\;patients\;in\;Gn}{\underset{m=1}{\sum }}\frac{THI\;score_{patient\;m}\;\left(T2\right)-THI\;score_{patient\;m}\;\left(T0\right)}{THI\;score_{patient\;m}\;\left(T0\right)}\;\;$$
where Gn ∈{G1, G2, G3, …, G10} representing each study group, and m ∈{1, 2, 3, …,  number of patients in Gn} representing each patient. A positive treatment outcome due to THI reduction from T0 to T2 is represented as a positive value of $TO_{15}$.

To evaluate the 15-day lasting therapeutic effect ($LTE_{15}$) after completely suspending all treatments, we calculated the average difference between the normalized THI scores immediately after 8 treatment sessions (T1) and normalized THI scores after 15 days of no treatment of any sort (T2). Normalization was performed in relation to the THI scores in the period T0 (before treatment) in all cases:(2)$$LTE_{15}\;of\;Gn=-\frac{1}{m}\times \overset{\#\;of\;patients\;in\;Gn}{\underset{m=1}{\sum }}\frac{THI\;score_{patient\;m}\;\left(T2\right)-THI\;score_{patient\;m}\;\left(T1\right)}{THI\;score_{patient\;m}\;\left(T0\right)}\;\;$$

In addition, we calculated the treatment outcome immediately after the finishing treatments ($TO_{0}$) to allow for comparison with the results of other studies only evaluating this outcome:(3)$$TO_{0}\;of\;Gn=-\frac{1}{m}\times \overset{\#\;of\;patients\;in\;Gn}{\underset{m=1}{\sum }}\frac{THI\;score_{patient\;m}\;\left(T1\right)-THI\;score_{patient\;m}\;\left(T0\right)}{THI\;score_{patient\;m}\;\left(T0\right)}\;\;$$

Since we are interested in knowing whether the effect of a treatment modality was superior to a placebo effect in most cases, we illustrated our results by subtracting the treatment outcome metrics $TO_{0}$ and $TO_{15}$ of the control (or placebo) group G1 from the treatment outcome metrics of each study group:$$THI\;reduction\;compared\;to\;control\;\left(0\;days\;after\;treatment\right)=TO_{0}\;of\;Gn-TO_{0}\;of\;G1$$
$$THI\;reduction\;compared\;to\;control\;\left(15\;days\;after\;treatment\right)=TO_{15}\;of\;Gn-TO_{15}\;of\;G1$$

Statistically significant differences between the THI scores of each study group were obtained by using Two-Way ANOVA analysis with the Tukey–Kramer as a post hoc test, whereas differences in THI score changes between T0 and T2 (normalized by the THI score at T0) was obtained by using Two-Way ANOVA analysis with Student–Newman–Keuls as a post hoc test. All statistical tests assumed normal distributions, checked by using the Kolmogorov–Smirnov normality test. The significance level considered for statistically significant differences (indicated by *, ** or by the *p*-value in figures) was 5% (i.e., *p* < 0.05 for statistically significant differences).

## 3. Results

Figure 4 shows the percentage reduction in THI scores after treatment. To eliminate potential influences due to the placebo effect, the percentage reduction in Figure 4 is shown after subtracting the THI score reduction by the negative control group (i.e., G1 with THI score reduction of −40.1%; Figure 4A). Original normalized values are G1 (−40.1%), G2 (−41%), G3 (−10.8%), G4 (−31.3%), G5 (−6.4%), G6 (−16.8%), G7 (−26.4%), G8 (−26.1%), G9 (−56.4%) and G10 (−44.7%). Significant differences were found only for the comparison between G5 and G9 (*p* < 0.05) and between G6 and G10 (*p* < 0.05). The highest THI score reduction was observed for LLLT auriculotherapy with a 15-min application time (G9), followed by laser puncture (G10) and LLLT combined with vacuum therapy (G2). Although the G2 THI score reduction (41%) was only slightly higher than the negative control (G1) group reduction (40.1%), a greater THI score reduction may potentially indicate that LLLT has a synergistic effect with vacuum therapy (G2). The opposite was found for all other study groups (Figure 4A). Vertix acted as an antagonist of LLLT combined with vacuum therapy. Even though Ginkgo biloba and LLLT had a synergistic effect with each other (G7 THI score reduction was greater than G6 and G8; Figure 4A), neither Ginkgo biloba nor LLLT showed positive treatment results compared with the negative control (G1; Figure 4A). Additionally, no positive treatment results have been shown by US and Vertix administration alone (Figure 4A).

In addition, the lasting therapeutic effect after 15 days of complete treatment suspension (i.e., $LTE_{15}$) was positive in all study groups except for G5 and G8 (Figure 4B). In fact, despite the final normalized THI scores being lower than those of the negative control group G1 (Figure 4A), the $LTE_{15}$ of G2, G4, G7, G9 and G10 were higher than the $LTE_{15}$ of G1 (Figure 4B). That means that THI scores may continue reducing more than the THI score reduction of the negative control group G1 and result in a long-term benefit to patients’ quality of life. The lasting therapeutic effect was especially pronounced for G10, which finished treatment with normalized THI scores below those of G1 at period T1 (Figure 4C; G1 normalized THI score = 38.0% and G10 normalized THI score = 33.6%). If THI scores had been compared immediately after eight treatment sessions (period T1, Figure 4C), G10 would have led to a lower THI score reduction compared with the negative control. When the lasting treatment effect is considered (Figure 4B), we could observe that normalized THI scores are higher than G1 (Figure 4A). Therefore, considering the lasting therapeutic effect should be part of future studies using the synergy between treatments based on this study, including longitudinal studies on the investigation of long-term treatment outcomes. Additionally, it is important to note that G9 and G10 not only resulted in positive treatment outcomes compared with the control group G1, but also induced higher 15-day lasting therapeutic effects of $LTE_{15}$, higher than G1 (Figure 4A,B).

Finally, our study suggests that the application time of LLLT auriculotherapy plays a key role in treatment outcome, as increasing the application time from 6 to 15 min not only meant producing a positive treatment outcome but also an increase of 39.6% for THI score reduction. This improvement can be seen by comparing the negative THI difference between G6 and the negative control group G1 (i.e., 16.8%–40.1% = −23.3%; Figure 4A) with the positive difference between G9 and G1 (i.e., 56.4%–40.1% = 16.3% THI score reduction; Figure 4A). It is worth noting that increasing the application time of LLLT auriculotherapy improved both the treatment outcome and lasting therapeutic effects. 

Figure 5 indicates the comparison between the initial × final values of THI scores for each treatment modality. Significant differences were observed in the initial × final ratio in groups G1 (*p* < 0.001), G2 (*p* < 0.01), G4 (*p* < 0.003), G5 (*p* < 0.05), G6 (*p* < 0.02), G7 (*p*< 0.05), G8 (*p* < 0.002), G9 (*p* < 0.0003) and G10 (*p* < 0.002). 

Figure 6 shows the number of patients at tinnitus grades from 1 to 5 for each study group based on the ranges of THI scores reported by Ferreira et al. [18]. The number of patients at tinnitus grade 1 is indicated by the bar on the left. Tinnitus grades increase from the left to the right and thus the number of patients at tinnitus grade 1 is indicated by the bar on the right of the group of columns of each group (illustrated by colors and the barplot X-axis. Figure 6A–C indicate how the number of patients at each tinnitus grade progresses from T0 to T1 to T2, respectively. We observed that our initial distribution of the number of patients is most often centered at around tinnitus grade 2 (initial THI scores from 18–36 at T0). As the treatment progresses, this distribution becomes centered at around tinnitus grade 1 (THI scores from 0–17) for treatment modalities, leading to the most positive outcomes, such as G9 and G10.

## 4. Discussion

### 4.1. Causes of Tinnitus and Treatment Modalities under Research

The relevance of our research relies on proposing either alternative or complementary treatments for tinnitus and subsequently improving patients’ quality of life. Tinnitus is a common otologic symptom defined as the conscious awareness of a sound in the absence of an external auditory stimulus [19]. Exposing the ears for a prolonged period to sounds above 85 decibels progressively causes damage, leading to hearing loss and sometimes causing dizziness and tinnitus. Tinnitus and hearing loss are frequently occurring comorbidities. Previous studies showed that 85 to 96% of patients with tinnitus have some degree of hearing loss and that only 8 to 10% have normal audiometry [20]. The most common comorbidities found in patients with hearing loss are associated with smoking, arterial hypertension (AH), diabetes mellitus (DM), lifestyle, aging, health history and leisure activities and occupational exposures [21,22]. The incidence of auditory symptoms is suggested to be correlated with lifetime exposure to noise. [21,22] 

The dysfunction theory concluded that the cochlea is damaged by loud noise, drug exposure or viral infections [23]. The pathophysiological mechanisms of tinnitus are not yet fully understood. However, there is currently a consensus that it is an abnormal neuronal activity in the inner ear or in the central auditory pathways, with an excitatory etiology (Figure 7). The etiology of tinnitus may be related to temporomandibular disorders (TMD), since correlation has been found [24], but explanations are not well defined yet. Several theories attempt to justify this correlation, such as the existence of an anatomical relationship between the hearing and the stomatognathic systems [24]. 

Since the cause of tinnitus is not well understood, treatment planning is often ineffective and subjective when treating idiopathic patients and patients who do not respond to conventional therapy (i.e., refractory patients). Therefore, developing personalized treatments that work for idiopathic and refractory patients are greatly needed. To treat these patients, low-level laser therapy (LLLT) has been chosen as one of the most promising candidates for tinnitus complementary and/or alternative treatments. This choice relies on several reasons, including (1) the current impossibility to determine the cause and find FDA-approved drugs for idiopathic and refractory tinnitus based on pharmacogenomics of hearing disorders [25], (2) no specific genetic locus strongly associated with tinnitus has been found despite the previous studies alluding to tinnitus genetic factors in tinnitus [26] and (3) treating tinnitus with drugs can cause systemic side effects on a short-term and/or long-term basis, including potentially more side effects than with LLLT applied locally on biological tissues around the auditory system and typically inducing photobiomodulation for up to 12 weeks after complete cessation of therapy [27]

In addition to LLLT, there are several modalities for the treatment of tinnitus, with emphasis on pharmaceutical therapy, physical therapy, psychotherapy and surgery, among others. All these modalities aim to reduce the intensity and annoyance that tinnitus causes to the individual. However, since its pathophysiology has not been well defined, there is no proven effective treatment to solve the problem in all cases. Among the possibilities of intervention, interest in the use of LLLT for photobiomodulation has been growing since a few decades ago. Research on the effect of LLLT on biological tissues began in the 1980s, although its development dates back to the 1960s [9,28]. Thus, international articles on this subject are found in the literature. 

### 4.2. Mechanisms of Action in LLLT

In general, understanding the mechanism of action of successful treatment modalities for a certain disease can narrow down potential treatment modalities that could synergize with the successful treatment. As discussed in the introduction section, we have proposed the application of vacuum therapy or US, hypothesizing that such an application could enhance the peripheral and regional blood circulation of the hearing system. Still, the photochemistry and photobiology of LLLT is complex due to multiple metabolic pathway interactions being possible. Physiological changes induced via LLLT occur due to photochemical mechanisms of laser therapy, in which chromophores absorb light and cause biochemical changes at immunologic and metabolic levels. The transmembrane protein complex cytochrome c oxidase (Cox) is the main component responsible for photon absorption, which leads to electronically excited states at the cellular level and faster electron transfer reactions. Increased electron transport potentially leads to increased ATP production in the tissues irradiated during LLLT. LLLT applies light to a biological system to promote tissue regeneration, reduce inflammation and relieve pain. Its inflammatory modulating action is obtained by accelerating microcirculation, which determines the types of changes in the hydrostatic pressure of capillaries, with absorption of edema and inactivation of intermediate catabolites. On the other hand, its analgesic action occurs through the inhibition and release by the central nervous system of pain mediators of endogenous analgesic substances, such as endorphins, which makes the transmission of the painful stimulus difficult. The molecular and cellular mechanisms of laser therapy suggest that this irradiation technique is capable of inducing biological processes using photon energy, which are subsequently absorbed by mitochondria. This stimulates the production of adenosine triphosphate (ATP) and low levels of reactive oxygen species (ROS), which activate transcription factors and induce many gene transcripts responsible for the beneficial effects of LLLT.

Repair mechanisms of LLLT derive from increased blood flow and activation of the mitochondria of the inner ear as well as hair cells, thereby stimulating the proliferation of inner ear cells and collagen production. Considering that laser devices for LLLT can focus the light beam in one direction and emit a specific wavelength, LLLT can be used locally to stimulate specific molecules through photobiomodulation of targeted tissues and cells. In general, the power of the devices currently available varies from 100 mW to 250 mW, and the wavelength varies from 600 to 850 nm, which seems to be sufficient to present positive results in the treatment of tinnitus. Possible mechanisms that explain the actions of laser therapy in the treatment of tinnitus include secretion of growth factors, increased cell proliferation and improved blood flow to the inner ear [29]. Despite LLLT being effective for numerous applications [27], previous studies have reported divergences regarding the application protocol and the effectiveness of LLLT for tinnitus [15,16].

The application of LLLT in the form of laser puncture stimulates the same biochemical, functional and neurological sequences as traditional needle acupuncture [30,31]. The needles are replaced by the energy delivered through the laser beam non-invasively and painlessly—especially advantageous for people who have a phobia of needles. The interaction between the laser and the points on the meridians of traditional Chinese medicine acupuncture is associated with the ability to store energy and generate electrical and heat stimuli in such points. The skin works as an optical membrane of radiation in the nearby integuments. Laser radiation emits a sufficient energy exchange to biologically active elements that regenerate neuronal electrical conductivity and restore the functional and energetic balance of illuminated tissues [31].

### 4.3. LLLT Specifications and Effectiveness

As in our study, the literature reports studies [9,28] with positive results for the application of LLLT to treat tinnitus. However, the effect of cochlear LLLT has only been investigated with laser power levels <7.5 mW. In our study, we showed that no significant effect has been observed upon increasing the power to 100 mW per spot for three spots illuminated with 660 nm and three spots illuminated with 808 nm, while combining LLLT with vacuum therapy with or without flunarizine dihydrochloride. In particular, this insignificant effect was assessed by no statistically significant difference being found in the THI total between the placebo (G1) and treated groups (G2 and G3).

Despite the lack of cochlear LLLT studies using laser power levels >7.5 mW in the literature, these levels are typically reported for transmeatal LLLT studies. One of the oldest examples of transmeatal LLLT studies is the research carried out by Shiomi et al. (1997) [32], in which 38 refractory patients who suffered from tinnitus (i.e., patients with no response to conventional therapies) were treated with 830-nm transmeatal LLLT irradiated via the external auditory canal towards the cochlea with 40 mW of power, once a week for approximately ten weeks. After the irradiations, the subjects scored the volume, duration and degree of tinnitus discomfort on a five-point scale. The authors concluded that although only 26% of the patients showed improvement in the evaluated criteria, the duration, volume and degree of annoyance showed improvement by up to 55% to 58%, demonstrating that photobiomodulation is a viable option to rehabilitate patients with intractable tinnitus [32]. 

Similarly, Mirz et al. [33], Rhee et al. [34] and Demirkol et al. [35] found statistically significant differences in scores of the visual analogue scale (VAS), tinnitus handicap inventory (THI) total and/or THI functional. Demirkol et al. [35] used illumination at 810 nm or 1064 nm at 250 mW of laser power, whereas Mirz et al. [33] and Rhee et al. [34] used illumination at 830 nm at 50 mW and 67 mW of laser power, respectively. Mirz et al. [33], Rhee et al. [34] and Demirkol et al. [35] observed these differences once tinnitus patients received 15 treatment sessions of 15 min over 3 weeks, 12 sessions of 20 min over 4 weeks and 10 sessions of 20 s/day (for the 1064 nm laser) or 9 s/day (for the 810 nm laser) over 2 weeks, respectively. Patients were 41 sensorineural hearing loss (SNHL) chronic unilateral or bilateral tinnitus patients, 50 unilateral or bilateral tinnitus patients and 46 subjective bilateral tinnitus and temporomandibular disorder patients, respectively for the studies of Mirz et al. [33], Rhee et al. [34] and Demirkol et al. [35].

Conversely, other studies have shown the ineffectiveness of LLLT for tinnitus in several conditions. By using lasers emitting at either 810 nm or 830 nm, and 60 mW, 100 mW and 450 mW of laser power, Nakashima et al. [36], Choi et al. [37] and Dejakum et al. [38] suggested that LLLT was ineffective. In the studies of Nakashima et al. [36], Choi et al. [37] and Dejakum et al. [38], tinnitus patients received 4 treatment sessions of 6 min over 4 weeks, 10 sessions of 20 min over 2 weeks, and 12 sessions of 30 min over 4 weeks, respectively. These patients were 45 sensorineural hearing loss (SNHL) patients, 38 SNHL and chronic unilateral tinnitus patients and 47 chronic tinnitus patients, respectively, in the studies of Nakashima et al. [36], Choi et al. [37] and Dejakum et al. [38].

In terms of LLLT for all types of tinnitus using similar laser power levels as our study, previous studies have reported results only for transmeatal LLLT with wavelengths of ≥810 nm (i.e., 810-nm, 830-nm or 1064-nm). In these studies, effective LLLT outcomes were only found for application times <10 min and no association could be retrieved from laser power levels (Nakashima et al. [36], Choi et al. [37] and Dejakum et al. [38] reported a statistically insignificant treatment outcome for power levels of 60 mW, 100 mW and 450 mW, whereas Mirz et al. [33], Rhee et al. [34] and Demirkol et al. [35] reported statistically significant treatment outcome for power levels of 50 mW, 67 mW and 250 mW). With this in mind, we expected that delivering light to the auditory system at similar laser power levels would lead to effective LLLT outcomes for times < 10 min. 

However, the above trend in the laser power levels has not been observed in our study (Figure 4), as G9 (15 min of irradiation time) led to a greater treatment outcome than G6 and G7 (6 min of irradiation time) when considering transmeatal LLLT with 100 mW power levels. This trend may have changed due to using the 660-nm wavelength. It is worth noting that 660 nm penetrates shallower than wavelengths ≥ 810 nm, and hence the power levels to the light delivered to the auditory system are smaller. Furthermore, Mirz et al. [33] only reported a significant statistical difference in THI total between the placebo and treated groups, which does not mean LLLT was effective in tinnitus patients, as the authors clearly stated “Tinnitus is not reduced by low-power laser treatment”. Section 4.4 is dedicated to comparing the treatment outcome of our study with previous studies using comparable LLLT parameters to treat tinnitus.

### 4.4. Comparison of Achieved Treatment Outcome of Transmeatal LLLT

With the above in mind, when comparing our study with the LLLT outcome of the 830-nm laser power levels used and same metrics/scales to evaluate this outcome (i.e., THI total), our study is more comparable with those of:Choi et al. [37]: for the placebo group, THI total of 48.4% ± 24.4% at T0, 44.2% ± 22.0% at T1 and 43.4% ± 20.4% at T2, and, for the treated group, 38.8% ± 25.4% at T0, 33.9% ± 29.1% at T1, 34.7% ± 28.2% at T2.Mirz et al. [33]: for the placebo group, THI total of 45.7% ± 19.9% at T0 and 38.7% ± 21.8% at two weeks after T2, i.e., 1 month after complete cessation of treatment, and, for the treated group, THI total of 39.8% ± 24.8% at T0 and 38.8% ± 24.1% at two weeks after T2.Rhee et al. [34]: for the placebo group, THI total of 54.6% ± 29.9% at T0 and 47.6% ± 27.4% between T1 and T2 (1 week after complete cessation of treatment), i.e., 1 month after complete cessation of treatment, and, for the treated group, THI total of 61.6% ± 24.8% at T0 and 48.9% ± 23.2% between T1 and T2.

If comparing studies solely based on statistically insignificant and significant changes in the THI total, our results for G6 agree with the insignificant LLLT outcome reported by Choi et al. [37], whereas the results for G9 agree with the LLLT outcome reported by Mirz et al. [33] and Rhee et al. [34] However, it is important to note that most previous studies did not show a positive treatment outcome based on the reduction in THI total over the duration of treatment. Since previous studies do not report the individual data of each patient to calculate $TO_{15}$ based on individual changes in the THI total, the average treatment outcome of a treated group n (Gn) can be calculated by using the reported mean THI total, i.e., by calculating the difference $-\frac{mean\;THI\;score\;\left(T2\right)-mean\;THI\;score\;\left(T0\right)}{mean\;THI\;score\;\left(T0\right)}$ between the mean THI total in T0 and T2, normalized by the mean THI total at T0, individually for the placebo group and with the same difference for the treated group. By performing such calculation, we obtain the average $TO_{15}$ subtracted from the placebo effect: $$Average\;TO_{15}=-\left(\frac{mean\;THI\;score_{treated}\;\left(T2\right)-mean\;THI\;score_{treated}\;\left(T0\right)}{mean\;THI\;score_{treated}\;\left(T0\right)}\;\frac{mean\;THI\;score_{placebo}\;\left(T2\right)-mean\;THI\;score_{placebo}\;\left(T0\right)}{mean\;THI\;score_{placebo}\;\left(T0\right)}\right)$$


In our study:

$$Average\;TO_{15}\;of\;G6=-\left(\frac{32.40\%-41.60\%}{41.60\%}-\frac{31.46\%-50.36\%}{50.36\%}\right)\approx 22.1\%-37.5\%\;=-15.4\%$$


$$Average\;TO_{15}\;of\;G9=-\left(\frac{16.55\%-35.64\%}{35.64\%}-\frac{31.46\%-50.36\%}{50.36\%}\right)\approx 53.4\%-37.5\%=15.9\%$$



In the study of Choi et al. [37]:


$$Average\;TO_{15}\;\left(Choi\;et.\;al.\right)=-\left(\frac{34.7\%-38.8\%}{38.8\%}-\frac{43.4\%-48.4\%}{48.4\%}\right)\approx 10.6\%-10.3\%=0.3\%$$


In the study of Mirz et al. [33]:


$$Average\;TO_{15}\;\left(Mirz\;et.\;al.\right)=-\left(\frac{38.8\%-39.8\%}{39.8\%}-\frac{38.7\%-45.7\%}{45.7\%}\right)\approx 2.5\%-15.3\%=-12.8\%$$


In the study of Rhee et al. [34]:


$$Average\;TO_{15}\;\left(Rhee\;et.\;al.\right)=-\left(\frac{48.9\%-61.6\%}{61.6\%}-\frac{47.6\%-54.6\%}{54.6\%}\right)\approx 20.6\%-12.8\%=-7.8\%$$


### 4.5. Novelty of this Study and Future Prospects

Considering the calculations above and that Choi et al. [37], Mirz et al. [33] and Rhee et al. [34] reported that LLLT of tinnitus was not significantly more effective than the placebo effect, to the best of our knowledge our study is the first to show a strongly positive treatment outcome of transmeatal LLLT based on THI scores. This outcome was achieved for G9 by using transmeatal LLLT at 660 nm with 100 mW of laser power for 8 sessions of 15 min over 4 weeks. Additionally, our results suggest that the outcome of G9 and G10 was sufficiently positive when compared to the placebo, i.e.:(4)$$TO_{15}\;of\;G9-TO_{15}\;of\;G1\;\left(placebo\right)=56.4\%\;-40.1\%=16.3\%$$
(5)$$TO_{15}\;of\;G10-TO_{15}\;of\;G1\;\left(placebo\right)=44.7\%-40.1\%=4.6\%$$

Therefore, transmeatal LLLT and laser puncture showed potentially promising treatment outcomes in our study and should be further validated in future studies involving more patients while using the same treatment protocols as the current study. In addition, new protocols should investigate the dosimetry of transmeatal LLLT, as the treatment outcome has been shown to be strongly dependent on the irradiation time (comparison between G6 and G9; Figure 4). Finally, based on the randomized controlled trials (RCTs) reviewed by Chen et al. [15], the four studies [28,33,34,39] investigating LLLT for tinnitus treatment of only idiopathic patients reported no LLLT effect superior to the placebo effect. Therefore, to the best of our knowledge, our study is the first to show LLLT protocols with positive treatment outcomes for idiopathic patients. 

Our findings may contribute to advancements in the personalized medicine of idiopathic and refractory tinnitus patients. In particular, we showed that transmeatal LLLT with the protocol applied to G9 as well as the laser puncture protocol applied to G10 has the potential to be an alternative treatment for tinnitus. It is worth considering monitoring the long-term effect of the protocols of this study, as our results have shown that the 15-day lasting therapeutic effect of LLLT ($LTE_{15};$ Figure 4B) on patients of the study groups G2, G4, G7, G9 and G10 was higher than the lasting therapeutic effect of the placebo group G1 at 15 days after complete suspension of all treatment. If such a lasting effect lasts more than 15 days, patients of study groups with a higher lasting effect may exhibit treatment effects superior to placebo effects. 

Since photobiomodulation has been reported to potentially last up to the period of 12 weeks after complete cessation of therapy [27], future studies could monitor LLLT effects up to such a period. In addition, LLLT using our developed protocols could be further validated for personalized medicine of other patient groups such as those with temporomandibular disorders (TMD), since tinnitus is a very frequent symptom in individuals with TMD [24]. Furthermore, TMD may benefit from the anti-inflammatory photobiomodulation induced via LLLT [35]. Hence, future studies developing effective LLLT protocols for the simultaneous treatment of tinnitus and TMD may be a promising approach for the personalized medicine of tinnitus + TMD patients.

Finally, our study is the first to investigate the synergistic and antagonistic effects of LLLT combined with vacuum therapy, ultrasound (US), Ginkgo biloba and flunarizine dihydrochloride. We also compared all results with the treatment outcomes with laser puncture and transmeatal LLLT alone at 660 nm with the irradiation times of 6 and 15 min. Our results indicated that combinations between LLLT and other treatment modalities lead to short-term antagonistic effects when evaluating the THI total immediately after treatment. Except for transcochlear LLLT + vacuum therapy (G2), these antagonistic effects persisted 15 days after treatment. A positive treatment outcome of LLLT may be found if the lasting therapeutic effect (LTE) increases in a long-term manner (>15 days) despite the antagonistic effects up to 15 days. The long-term effects of LLLT on THI total remain unknown [24].

### 4.6. Limitations of the Study and Future Prospects

Our study has limitations associated with randomized control trials (RCTs). Due to the nature of randomized patient recruitment, patients with similar cofounding factors for treatment (e.g., age, gender and hearing loss) could not be selected to be part of the same study group. Homogenizing categories and quantities of all cofounding factors among study groups would require sampling a large patient population. Our study did not sample a sufficient number of patients to show confirmatory statistics representative of the entire Brazilian population. However, our results can be used to draw conclusions based on statistical tests similar to previous studies, which involved a similar number of patients compared to our study. Furthermore, our results allow for a comparison of the treatment outcome and long-term treatment effects by both complementing systematic reviews and extending the knowledge of the research community on synergistic and antagonistic of LLLT combined with vacuum therapy, ultrasound (US), Ginkgo biloba and flunarizine di-hydrochloride.

We emphasize that our results are the first steps in investigating the outcome of the most successful treatments for tinnitus in a larger population, thereby minimizing trends of treatment outcome generated by cofounding factors. With this in mind, future studies involving larger sample sizes can be performed, with protocols used for G2, G9 and G10. These studies should take into consideration the potential effects of conditions associated with tinnitus on the treatment outcome. In addition to the association between tinnitus and temporomandibular disorders, studies supporting the lack of peripheral irrigation in the inner ear as the cause of tinnitus also associate this lack as the cause of vertigo and hearing loss. In this case, sensorineural hearing loss caused by microcirculatory insufficiency may result from vascular occlusion due to embolism, hemorrhage or vasospasm, diabetes mellitus (DM) or arterial hypertension (AH). [4] It is worth noting that DM is the condition most commonly associated with hearing disorders compared with glucose-metabolic-related disorders [4]. Additionally, DM patients often present symptoms such as dizziness, tinnitus and hearing loss [4]. Therefore, monitoring conditions that can (directly or indirectly) indicate microcirculatory insufficiency in the inner ear could potentially lead to a better patient stratification prior to recruitment in future studies.

## 5. Conclusions

Our study has shown that LLLT was effective for the treatment of tinnitus. The most successful treatments were vacuum therapy + transcochlear LLLT (G2), transmeatal LLLT at a 660-nm wavelength and 100 mW of laser power for 15 min (G9) and laser puncture (G10). New protocols in studies involving a larger patient population must be carried out to refine the LLLT protocols proposed in our study. Future studies should investigate the long-term effects of LLLT in tinnitus patients, as well as the dosimetry, wavelength and number of sessions of LLLT. For future reference on designing new protocols based on our study, we showed:

1.a positive treatment outcome $\left(placebo\;TO_{15}\;40.1\%\right)$ by using either transmeatal LLLT alone at 660 nm with 100 mW of laser power for 8 sessions of 15 min over 4 weeks ($TO_{15}\;of\;G9=56.4\%$), or laser puncture with 4J in a total of 23 acupoints $\left(TO_{15}\;of\;G10=44.7\%\right)$,2.an improvement in the treatment outcome of transmeatal LLLT at 660 nm with 100 mW of laser power by increasing the irradiation time from 6 min $\left(TO_{15}\;of\;G6=16.8\%\right)$ to 15 min $\left(TO_{15}\;of\;G9=56.4\%\right)$,3.short-term antagonistic effects of vacuum therapy, ultrasound (US), Ginkgo biloba and flunarizine dihydrochloride when combined with LLLT $(TO_{0}\leq 35.4\%\;\mathrm{and}\;TO_{15}\leq \;31.3\%\;\mathrm{from}\;G2\;\mathrm{to}\;G8,\;$except for G2 at 15 days after complete cessation of treatment; $TO_{15}\;of\;G2=41.0\%)$, 4.a positive lasting therapeutic effect at 15 days after completely suspending all treatments ($placebo\;LTE_{15}=2.13\%$) upon combining LLLT with vacuum therapy (G2), LLLT with Ginkgo biloba (G7) or by using flunarizine dihydrochloride alone (G4), by using the transmeatal LLLT alone (G9) or by using laser puncture (G10) $(all\;LTE_{15}>4.37\%$).

## Figures and Tables

**Figure 1 jpm-13-00581-f001:**
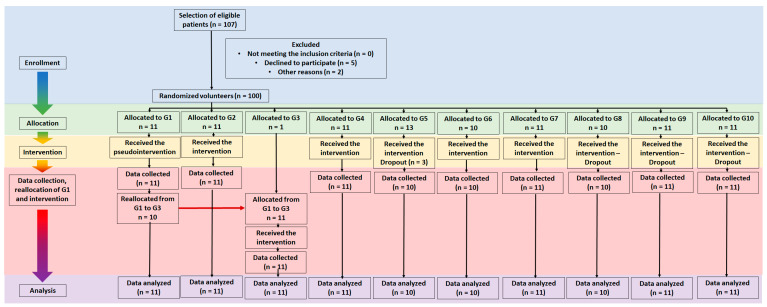
Flowchart of the steps taken throughout this study (CONSORT flow diagram). The flowchart highlights the enrollment of volunteers, their allocation, interventions applied, data collection, reallocation of G1 patients into G3 for subsequent G3 intervention and analysis during the study. Clinical evaluation was carried out in the pre- (T0) and post-treatment period (T1; after 8th session) and 15 days after the treatment has been completely suspended (T2). G1, G2, G3, …, G10 are the study groups from 1 to 10 described in Section 2.2 Study protocol.

**Figure 2 jpm-13-00581-f002:**
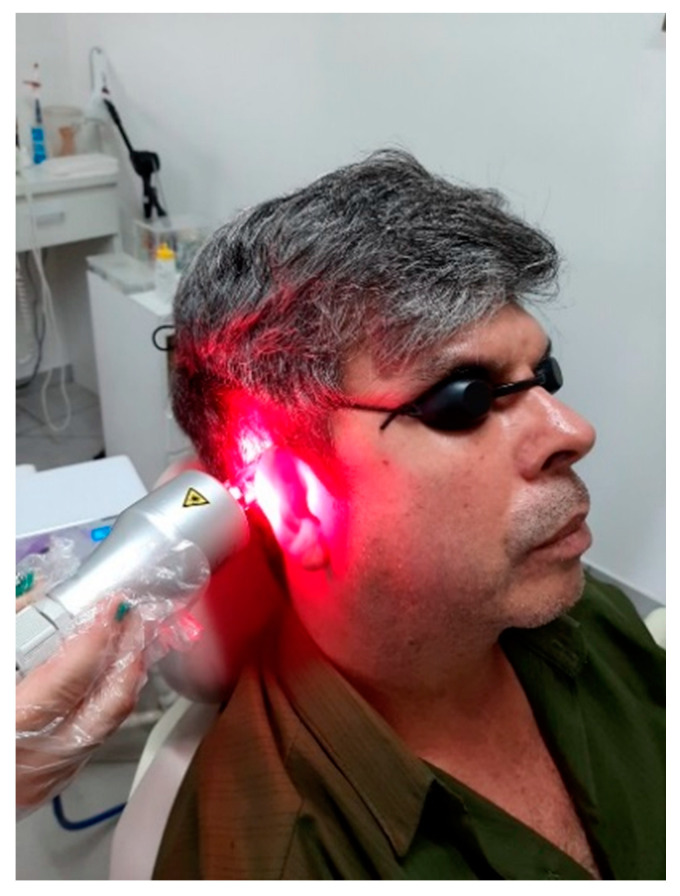
Device with suction cup and negative pressure combined with a cluster containing six LLLT outlets, three in red and three in infrared (VACUUMLASER, MMOPTICS, São Carlos–SP).

**Figure 3 jpm-13-00581-f003:**
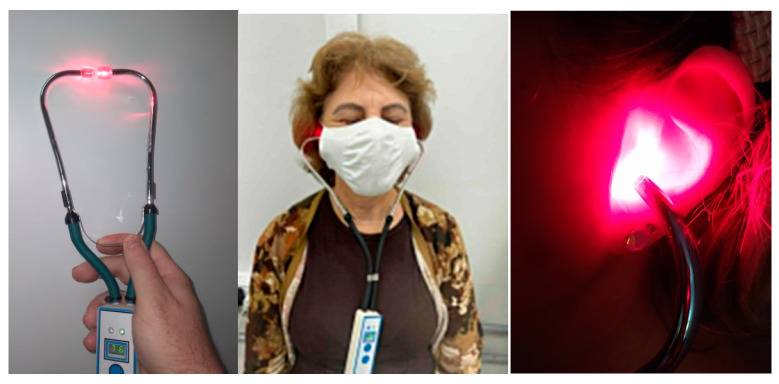
LLLT prototype device applying red laser with a wavelength of 660 nm developed at the Technological Support Laboratory of IFSC-USP for laser auriculotherapy.

**Figure 4 jpm-13-00581-f004:**
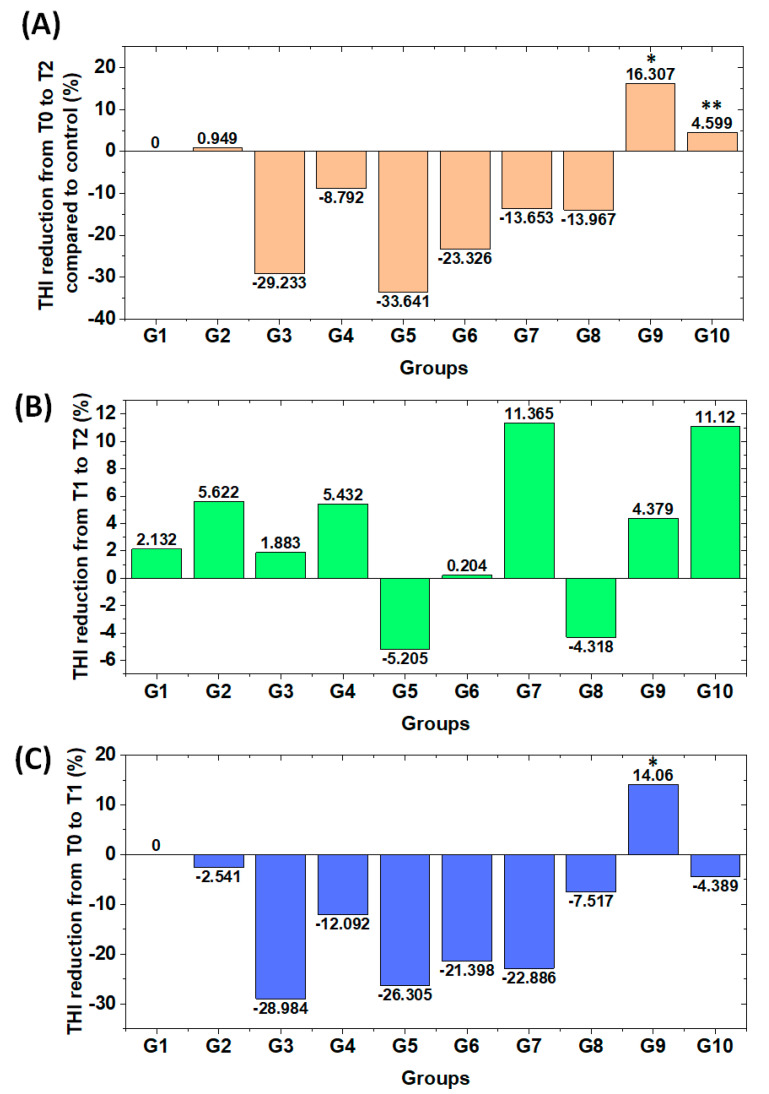
Comparison between study groups showing the evolution of different interventions to treat tinnitus. This comparison was made in relation to the (**A**) $TO_{15}$, i.e., average normalized THI score reduction 15 days after complete treatment suspension (period T2) and excluding the placebo effect (subtraction of the 40.1% THI score reduction induced in patients of G1 at T2); (**B**) $LTE_{15}$, i.e., average THI score reduction 15 days after the treatment has been completely suspended (i.e., the difference between the normalized THI scores immediately after 8 treatment sessions (period T1) and normalized THI scores 15 days of no treatment of any sort (period T2)) and (**C**) $TO_{0}$, i.e., average normalized THI score reduction immediately after the 8 treatment sessions (period T1) excluding the placebo effect (subtraction of the 38.0% THI score reduction induced in patients of G1 at T1). Statistically significant differences were observed between G5 × G9 (*p* < 0.05; indicated by * in Figure 4A,C) and G6 × G10 (*p* < 0.05; indicated by ** in Figure 4A).

**Figure 5 jpm-13-00581-f005:**
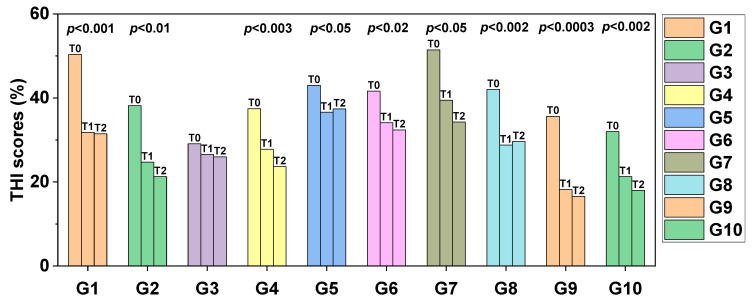
Barplot comparing the non-normalized THI scores at T0, T1 and T2 between tinnitus treatment modalities explored in this study (from G1 to G10). The left bar of each group represents initial THI scores before any treatment (period T0). The middle bar of each group represents scores after 8 treatment sessions (period T1). The right bar of each group represents the final THI scores 15 days after completely suspending treatment of any sort (period T2). Statistically significant differences between T2 and T0 were observed for all groups except for G3. The ballpark of *p*-values obtained can be seen in the top of the bars of each group.

**Figure 6 jpm-13-00581-f006:**
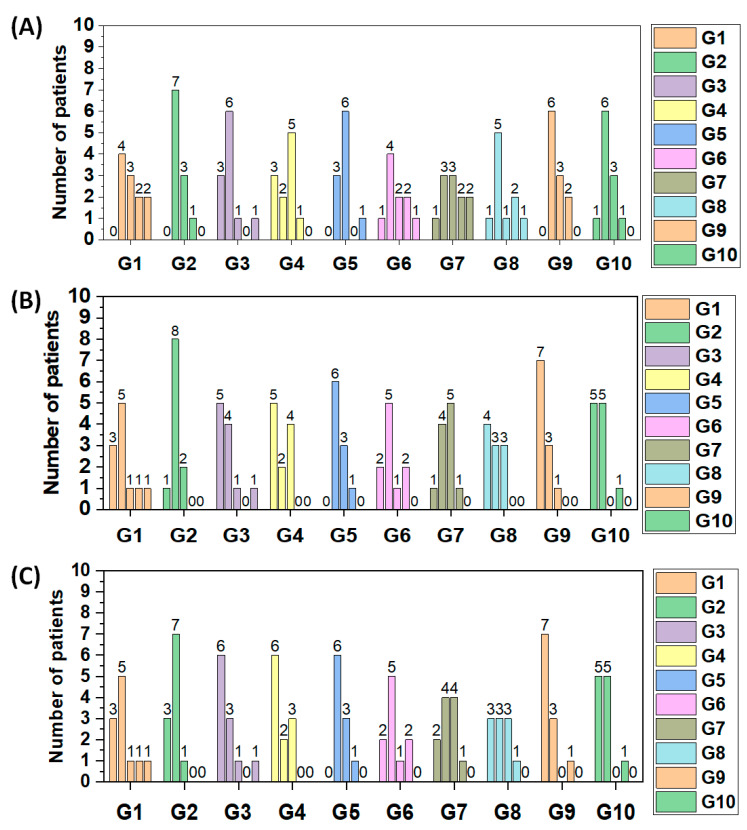
Evolution of the number of patients at each tinnitus grade after each period of each treatment. Barplot comparing the grades of tinnitus at (**A**) T0, (**B**) T1 and (**C**) T2 between tinnitus treatment modalities explored in this study (from G1 to G10). The bars from left to right indicate the number of patients from grade 1 to grade 5, respectively. Grade 1: THI scores from 0–17. Grade 2: THI scores from 18–36. Grade 3: THI scores from 37–56. Grade 4: THI scores from 57–76. Grade 5: THI scores from 77–100. The assessment and division of grades based on THI scores was based on the work of Ferreira et al. [18].

**Figure 7 jpm-13-00581-f007:**
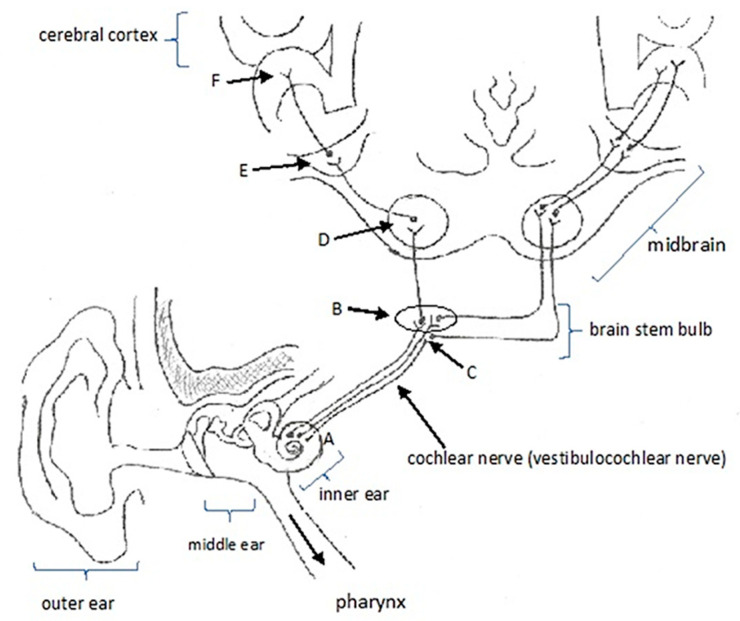
Schematic representation of the auditory pathway through the vestibulocochlear nerve (VIII cranial nerve) showing the path of the auditory pathway from the cochlear nerve which starts from the spiral ganglion of the cochlea (A) located in the inner ear. Next, the cochlear nerve goes to the superior portion of the brainstem bulb, where this nerve follows the homolateral path through the synapse in the dorsal cochlear nucleus (B) and follows the contralateral path through the synapse in the ventral cochlear nucleus (C). Then, the nerve continues with synapses in the inferior colliculus (D) and medial geniculate nuclei (E) (midbrain) and another synapse in the anterior transverse temporal gyrus (F) of the cerebral cortex.

**Table 1 jpm-13-00581-t001:** Patient demographics for all study groups and per study group.

Study Groups	Gender (Female:Male)	Age (Mean ± Standard Deviation)	Age (Range)
**All groups**	50:57	49.9 ± 8.2	28–63
**G1**	6:5	49.5 ± 9.1	37–63
**G2**	6:5	48.4 ± 6.2	36–57
**G3**	5:6	49.0 ± 8.4	37–60
**G4**	5:6	48.0 ± 9.5	28–60
**G5**	5:5	45.0 ± 9.7	29–60
**G6**	3:7	47.6 ± 8.2	38–60
**G7**	4:7	52.5 ± 8.5	37–60
**G8**	6:4	54.6 ± 5.9	42–60
**G9**	4:7	53.0 ± 7.4	39–60
**G10**	6:5	50.9 ± 6.8	37–60

## Data Availability

The data presented in this study are available on reasonable request from the corresponding author.

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
