# Peer review of "Effects of Red and Infrared Laser Therapy in Patients with Tinnitus: A Double-Blind, Clinical, Randomized Controlled Study Combining Light with Ultrasound, Drugs and Vacuum Therapy"

_jpm, 2023, doi:10.3390/jpm13040581_

Round 1
Reviewer 1 Report
Abstract:
The authors must explain the acronym LLLT in the abstract.
Introduction:
The authors should shorten it, avoiding to explain the results they are going to show. Remove lines 128-133.
Also, paragraphs (lines 125-153) on vacuum therapy and ultrasound should be shorten and moved before the aims of the stud.
Mat&Methods
With only 107 eligible patients, the authors should not make so many groups in comparison, being most on 10 patients.
It is very surprising their statistics... Normal distribution with 10 patients? Please explain or show the data.
Statistically significant differences between THI scores of each study group were 315 obtained by using Two-Way ANOVA analysis with the Tukey–Kramer as post-hoc test, 316 whereas differences between THI score changes between T0 and T2 (normalized by the 317 THI score at T0) was obtained by using Two-Way ANOVA analysis with Student–New-318 man–Keuls as post-hoc test. All statistical tests assumed normal distributions checked by 319 using the Kolmogorov -Smirnov normality test. The significance level considered for sta-320 tistical significant differences to (indicated by *, ** or by the p-value in figures) was 5% 321 (i.e., p < 0.05 for statistically significant differences).
Conclusions
They should be re-written avoding repeating results and only summing them.
Reviewer 2 Report
This manuscript investigates the effectiveness of potential alternative and complementary treatment modalities for tinnitus. The topic of this manuscript looks promising. However, there are major flaws in this paper. The methodology of the manuscript is confusing. The flow chart shown in figure 1 won’t align with the text in the methodology. In the text, it seems like the participants from group 1 were relocated to G2; on the other hand, the flow chart indicates something else. The results sections look incomplete, and I will recommend authors to add tables of two-way ANOVA analysis. There are numerous grammatical mistakes and inaccurate sentence structures throughout this paper, which affects the flow and readability. I will recommend authors to edit the manuscript thoroughly. The first subsections under the discussion section need to be elaborated. Section 4.3 is incohesive, and particularly for this section, I will recommend authors to avoid explaining the methods and results of 3-4 previous studies simultaneously in one sentence. The authors have not mentioned the limitations of this study adequately. The sample size of each group is very small. Thus it is important to write a separate section on the limitations of the study.
Reviewer 3 Report
This study investigated the effects of LLLT in patients with tinnitus using THI in a RCT. It is original to present a significant treatment option for tinnitus based on a RCT. However, the manuscript needs several modifications.
1. First of all, as authors mentioned, hearing loss is frequently accompanied by tinnitus. Hearing thresholds should be compared among groups (G1-G10) and presented. Differences in hearing threshold among groups could be confounding factors of treatment outcome.
2. Distribution of sex and age in each group should also be presented. And differences in sex and age among groups should be presented because these could also be confounding factors.
3. In addition, the number of 10 or 11 in each group seems to be insufficient to compare the outcome among groups. The small number of included patients in each group should be presented as the limitation of this study.
4. In Figure 1, ‘reallocation’ should be explained. In lines 214-252, the reallocation was not described. Is ‘Reallocated from G2 to G3’ correct? ‘Reallocated from G1 to G3’ seems correct. Check the flowchart and explain the reallocation in the manuscript.
5. The mechanism of effective treatment for tinnitus of auricle laserpuncture, which was a significant alternative treatment in this study, should be described in Discussion.
6. In line 25, explain the abbreviation of LLLT in Abstract.
7. In line 306, interest -> interested.
8. In lines 440 and 447, Figure 6 -> Figure 7.
Reviewer 4 Report
The study is multifaceted and I would have tested fewer treatments on larger groups of subjects. But it is still well structured.
The results are interesting. The discussion and conclusions are adequate. Undoubtedly it is necessary to calibrate the treatments with the identification of the optimal parameters of dosimetry, wavelength and number of sessions of LLLT.
Round 2
Reviewer 1 Report
All suggestions have been followed.
Reviewer 3 Report
The manuscript has been improved after authors’ revision.